# Concurrent and Longitudinal Predictors of Adolescent Delinquency in Mainland Chinese Adolescents: The Role of Materialism and Egocentrism

**DOI:** 10.3390/ijerph17207662

**Published:** 2020-10-21

**Authors:** Daniel T. L. Shek, Xiang Li, Xiaoqin Zhu, Esther Y. W. Shek

**Affiliations:** Department of Applied Social Sciences, The Hong Kong Polytechnic University, Hong Kong; xann.li@polyu.edu.hk (X.L.); xiaoqin.zhu@polyu.edu.hk (X.Z.); catowlshek@yahoo.com.hk (E.Y.W.S.)

**Keywords:** materialism, egocentrism, delinquency, adolescents, longitudinal design

## Abstract

Background: Although studies have examined the influence of materialism on adolescent well-being, there are several methodological limitations: studies examining the influence of materialism on adolescent delinquency are almost non-existent; researchers commonly used cross-sectional designs; the sample size in some studies was not large; validated measures on materialism in non-Western contexts are rare; there are very few Chinese studies. Besides, no study has examined the hypothesis that egocentrism is the mediator in the influence of materialism on adolescent delinquency. Methods: Using a short-term longitudinal design, two waves of data were collected from 2648 early adolescents in mainland China. At each wave, students completed validated measures of materialism, egocentrism and delinquent behavior. Results: Materialism and egocentrism positively predicted adolescent delinquency at Wave 1 and Wave 2 and over time. While materialism at Wave 1 positively predicted increase in delinquency over time, egocentrism did not. However, PROCESS analysis showed that egocentrism mediated the longitudinal influence of materialism on adolescent delinquent behavior. Conclusions: Materialism and egocentrism are predictors of adolescent delinquency, with egocentrism serving as a mediator in the influence of materialism on adolescent delinquency.

## 1. Introduction

In his book entitled “An Inquiry into the Nature and Causes of the Wealth of Nations” (The Wealth of Nations), Adam Smith [1] examined how a nation can build up its wealth. Coupled with the first Industrial Revolution taking place at almost the same time, accumulation of wealth, profit maximization and consumption have become the hallmark of capitalism. In the past few centuries, economic development through production of goods, material consumption and wealth accumulation has been common in many countries. With intensive globalization in recent decades, the number of investment banks, stock markets and bond markets has increased tremendously, focusing on wealth creation and asset accumulation. In many social indicator systems, economic growth and income are also important indicators of social progress. For example, the Human Development Index employs GDP as an indicator of social progress; the Organization for Economic Co-operation and Development (OECD) Better Life Index also uses income and wealth as social indicators. The OECD is an international organization that provides a platform to share policies and experiences that can promote better lives.

The focus on accumulation of wealth, consumption of goods and making money is closely related to the concept of “materialism”, which is commonly understood as over-emphasis on material possessions such as money. In the scientific literature, there are different conceptions of materialism in terms of values, needs and motives [2,3]. In a systematic review of the materialism literature, Kasser [2] regarded materialism as comprising “a set of values and goals focused on wealth, possessions, image, and status. These aims are a fundamental aspect of the human value” (p. 489). In the early work on materialism, Richins and Dawson [4] highlighted three attributes of materialism, including placing possessions at the center of one’s life, considering possessions and their acquisition essential to happiness and life satisfaction and defining success in terms of possessions. In a systematic review of how materialism is linked to well-being, Dittmar, Bond, Hurst and Kasser [5] defined materialism as “people’s long-term endorsement of values, goals, and associated beliefs that center on the importance of acquiring money and possessions that convey status” (p. 880). In this paper, we conceived materialism as beliefs about the centrality of acquiring material possessions, the importance of material possession and hedonistic pursuits. Culturally speaking, materialism is quite deep-rooted in contemporary societies. In 1976, a Swedish pop group (ABBA) released a song entitled “Money, Money, Money”. This song was wrote and produced by two members from ABBA (Benny Andersson & Björn Ulvaeus). Part of the lyrics is: “Money, money, money, must be funny in the rich man’s world. Money, money, money, always sunny in the rich man’s world”. Ironically, although Catholic and Christian values are upheld in many developed Western countries, such cultural emphasis on materialism is in sharp contrast to the Biblical teaching that “For the love of money is a root of all kinds of evil. Some people, eager for money, have wandered from the faith and pierced themselves with many griefs” (Timothy I: Chapter 6, verse 10).

What is the influence of materialism on human well-being? Based on 259 studies regarding how materialism is associated with personal well-being, Dittmar et al. [5] identified several interesting phenomena. First, although there was an overall negative relationship between materialism and well-being, there were variations in the strength of relationships across different indicators of materialism and different dimensions of well-being. Second, the findings revealed some moderators (such as socio-demographic factors) of the relationship. Third, there are few studies examining mediators of the relationship between materialism and well-being. According to Shrum et al. [3], although materialism “has a generally held connotation that is associated with character deficiencies, self-centeredness, and unhappiness, and most extant research views materialism as having a negative influence on well-being” (p. 1), it may have “good”, “bad” and “ugly” consequences. Regarding the “good” consequences of materialism, they included coping with self-threat and short-term utility. For the “bad” consequences of materialism, they included impaired physical health, negative mood, mental problems and pathological consumption behavior. Finally, people generally perceive individuals with materialistic behaviors in an “ugly” manner with undesirable stereotypes. However, Shrum et al. [3] warned that the research studies are inconclusive because of the dominance of cross-sectional research studies in the field. Kasser [2] outlined eight issues for future research, including the development of implicit measures of materialism, implementation of longitudinal studies based on children and vulnerable groups, examination of materialism and problematic outcomes, continued materialism despite low satisfaction, increasing external validity of studies, reduction of materialism, policies that decrease materialism and maintenance of a healthy view towards material possession.

The literature review shows several methodological problems in the studies on the relationship between materialism and well-being. The first limitation is that there are few studies based on adolescents, although adolescence coincides with a substantial increase in materialism [6]. In the review study of Dittmar et al. [5], 222 studies employed participants over 18 years, 31 studies recruited participants 18 years and under and 5 studies included both adults and non-adults. There are several reasons why researchers should conduct more related studies based on adolescents. First, adolescents begin to have more opportunities to use money when they enter high school (e.g., money to buy lunch) and peer comparison on possessing trendy materials (such as sports shoes and mobile phones) is very common. Second, with cognitive maturation, adolescents begin to explore materialistic values (e.g., earning money to enjoy life) and non-materialistic values (e.g., quest for a meaningful life and eternity) from different perspectives. Third, some adolescent developmental issues such as delinquency (e.g., compensated dating, shoplifting and earning quick money via illegal means) are closely related to materialism. In his review of the studies on materialism, Kasser [2] explicitly recommended that “substantially more research on materialism is needed that utilizes samples of children …” (p. 507).

The second methodological limitation is that existing studies are mostly Western studies and Chinese studies are almost non-existent. Amongst the 259 studies in the review conducted by Dittmar et al. [5], there were only 22 Asian studies and only three of them recruited Chinese samples. There are several reasons why we need Chinese studies. First, in terms of population size, Chinese people constitute roughly 18.4% of the world’s population [7]. Hence, if we want to generalize Western theories and findings worldwide, we need to recruit Chinese participants. Second, different from Western culture, Chinese culture is more collectivistic, focusing more on common interests than individual interests. Findings obtained based on the Western population may not be equally applicable to the Chinese context. Third, materialism may have different meanings in Chinese culture. In Chinese mainstream social philosophies, there is not much emphasis on money and material possession. For example, in Confucian thoughts, the focus is on self-cultivation of non-material virtues and money is regulated by li (propriety), as exemplified by the saying that “*jun zi ai cai, qu zhi you dao*” (gentlemen love fortune but in a proper way). In Buddhist thoughts, the emphasis is the transcendence of the material world as reflected in the saying of “*kan po hong chen*” (looking beyond the material world). On the other hand, there is a strong emphasis on materialism in Chinese folk culture. For example, while the concept of “*feng yi zu shi*” (have ample food and clothing) was proposed more than two thousand years ago, it is still a dominant Chinese belief. One clear indication is that people commonly say “*gong xi fa cai*” (may you make much money) to each other during Lunar Chinese New Year.

The third limitation surrounds the assessment of materialism. As Kasser [2] commented, “although diversity of measurement has advantages, many studies use single-item measures or measures with unknown or questionable psychometric properties” (p. 493). For an objective measure of materialism, it should have good reliability and validity (particularly factorial validity if there are different dimensions). While it is possible to have translated measures of materialism, it would be desirable to develop locally validated measures that adapt to Chinese culture to accurately capture materialism in the non-Western context.

The final limitation is that most studies in this field are cross-sectional studies with some of them having small samples. In his systematic review on materialism and personal well-being, Dittmar et al. [5] reported that there were 235 cross-sectional studies, 5 comparison studies and only 18 longitudinal studies in the field. While cross-sectional studies are cheap and easy to conduct, it is not possible to establish a causal relationship between the predictors and criterion variables because both of them occur at the same time. Alternatively, longitudinal studies have the advantage of looking at the predictor–criterion relationship over time, although their cost is high and following up of the participants is not easy. Obviously, if we want to look at the influence of materialism on adolescent well-being, longitudinal data would give a better answer to the question. As suggested by Kasser [2], “substantially more research on materialism is needed … uses prospective, longitudinal, and experimental designs” (p. 507). Besides, sample sizes in some of the cross-sectional studies were small: Kwak, Zinkhan and French [8] used 76 participants; Noris, Lambert, DeWall and Fincham [9] employed 61 participants. In their meta-analytic study, Dittmar et al. [5] reported that the medium sample size in the studies was 207. Hence, employing a large sample size is a desired methodological arrangement in future studies.

In addition to the above-mentioned methodological limitations, there are two conceptual issues intrinsic to the existing studies on how materialism is linked to well-being. The first conceptual issue is the definition of well-being and the related indicators. According to Dittmar et al.’s review [5], well-being is a broad concept that covers individual experience in a wide range of domains, such as self, emotion, health and behavior. Indeed, different indicators have been used to measure personal well-being, including subjective well-being (satisfaction with life, positive emotion and negative emotion), self-appraisals (positive self and negative self), Diagnostic and Statistical Manual Axis 1 mental disorders (anxiety, depression and compulsive buying), physical health and physical risk behaviors (e.g., delinquent behavior). Surprisingly, very few studies have examined the linkage between materialism and well-being manifested in behaviors such as antisocial behavior, particularly adolescent delinquency.

There are three reasons why we should examine materialism and antisocial behavior, particularly in adolescents. First, researchers have regarded some delinquent behavior (such as vandalism and theft) as externalizing behavior [10,11], which is commonly used as an indicator of well-being. Second, many forms of adolescent delinquent behavior are motivated by materialistic incentives, such as compensated dating, selling pirated items and shoplifting. Third, according to Donnelly et al. [9], individuals with materialistic values may suffer from cognitive deconstruction and aversive self-awareness, resulting in negative emotions and irrational behaviors. As delinquent behavior is irrational and impulsive, it is reasonable to expect that materialism is positively associated with delinquency. Nevertheless, only a handful of studies have provided support for this proposed positive relationship: Bilsky and Hermann [12] showed that hedonism was positively related to adolescent delinquency; Froggio and Lori [13] showed that hedonistic and materialistic values were positively associated with deviance in a “modest” manner; Auerbach et al. [14] revealed that materialism predicted greater risky behavior engagement (e.g., aggression and delinquency) in Chinese adolescents. There are also findings revealing that materialism is negatively linked to prosocial behavior. Yang, Fu, Yu and Lv [15] showed that materialism predicted decline in prosocial behavior to friends and strangers (not family members) and suggested that delinquency was an outcome instead of a precursor of materialism. As commented by Bilsky and Hermann [12], “the systematic relation between delinquency and the overall spectrum of individual values has not been a particular topic of psychological research in the past” (p. 921).

The second conceptual issue is on the mediating processes involved in the influence of materialism on adolescent well-being. Dittmar et al. [5] remarked that few studies have examined mediators regarding the influence of materialism on well-being. In different studies, materialism has been found to be correlated with different aspects of self. For example, Briggs, Landry and Wood [16] showed that individual factors (attitude, self-esteem and materialism) influenced attitude towards a volunteering task which eventually predicted helping behaviors. Based on a small sample of university students and guided by the empty self-theory and absorption–addiction theory, Reeve, Baker and Truluck [17] showed that materialism and compulsive buying were associated with celebrity worship, and these three factors were negatively associated with clarity of self-concept and well-being. Noguti and Bokeyar [18] found support for the hypothesis that low self-concept clarity was associated with a higher materialism level, particularly in women. Based on four studies, Jiang et al. [19] further reported that peer rejection impaired implicit self-esteem which further enhanced adolescent materialism, with implicit self-esteem serving as a mediator in the peer rejection–materialism relationship. The findings suggested that people with low self-esteem would engage in materialistic behaviors to compensate impaired self-esteem.

Besides, Donnelly et al. [6] argued that materialism is associated with one’s cognition, affect and behaviors through a six-step model entitled “escape of the self”, including falling short of one’s standard and self-disappointment, self-blaming, high aversive experience of oneself, presence of dysphoric moods, cognitive deconstruction (narrow and rigid thinking which helps one to escape from negative emotions) and impulsive behavior which is short-sighted and not rational in nature. Based on this model, they argued that materialism can be seen as a means to “escape” from the negative emotions in the first four steps and to form a new self. Furthermore, in the review of the influence of materialism on well-being, Dittmar et al. [5] used positive self-appraisals (i.e., positive self-evaluation such as self-esteem and self-actualization) and negative self-appraisals (i.e., negative self-evaluation such as self-doubt, self-discrepancies and self-ambivalence) to summarize the findings on the impact of materialism on well-being. The general observation is that materialism reduces positive appraisals but increases negative self-appraisals.

Based on the above discussion, a materialistic individual is likely to have low self-esteem, which may trigger and be reinforced by materialism. According to Awanis, Schlegelmilch and Cui [20], materialistic people “are seen as self-centered individuals who prefer to build meaningful relationships with possessions rather than people” (p. 183). Taken together, it is possible that the self-centered intention (i.e., egocentrism) of a materialistic individual functions as a compensation of one’s low self-esteem. Based on this reasoning, we argue that egocentrism is likely to take place under cognitive deconstruction and as a reaction to low self-esteem for three reasons. First, egocentrism (such as the perception of self-importance and self-conceit) can reduce negative emotions associated with materialism. Second, we can regard egocentrism as a defense mechanism to cope with the negative experience of self-inadequacy by self-inflation (i.e., Freudian defense mechanism of “reaction formation”). Third, there are some studies showing a positive relationship between materialism and narcissism [21,22] and a negative relationship between materialism and empathy [23].

While materialism may predict ecocentrism, there is also direct and indirect evidence showing a positive association between egocentrism and adolescent delinquency. For example, Chandler [24] showed that egocentrism was positively related to antisocial behavior. Greene et al. [25] showed that sensation seeking and egocentrism predicted risk-taking behaviors in adolescents. Górnik-Durose [21] showed that narcissism (a concept closely related to egocentrism) served as a mediator between materialism and well-being indicated by emotional well-being and psychological well-being. In conjunction with the above-mentioned findings on the relationship between materialism and egocentrism, we proposed that egocentrism may serve as a mediator of the influence of materialism on adolescent delinquency.

In view of the above conceptual and methodological gaps, we examined the concurrent and longitudinal predictive effects of materialism and egocentrism on adolescent delinquency using a short-term longitudinal study in which Chinese mainland high school students responded to validated measures of materialism, egocentrism and delinquency at two time points separated by one year. Specifically, we attempted to address four research questions as follows:

Research Question 1: Does materialism predict adolescent delinquency concurrently and longitudinally? We expected that materialism would have a concurrent and longitudinal positive prediction on adolescent delinquency (Hypotheses 1a and 1b).

Research Question 2: Does egocentrism predict adolescent delinquency concurrently and longitudinally? In accordance with the scientific literature, we expected that egocentrism would have a concurrent and longitudinal positive prediction on adolescent delinquency (Hypotheses 2a and 2b).

Research Question 3: Does materialism predict egocentrism at the same time and over time? Based on the notion of cognitive deconstruction and the above discussion, we expected that materialism positively predicted egocentrism at the same time and over time (Hypotheses 3a and 3b).

Research Question 4: Does egocentrism mediate the influence of materialism on adolescent delinquency? Based on the above discussion, we expected that egocentrism would mediate the influence of materialism on adolescent delinquency (Hypothesis 4).

## 2. Materials and Methods

### 2.1. Participants and Procedures

A longitudinal research design was used in this study, with data collected at two time points with one year apart. We recruited four high schools in mainland China from the schools participating in the “Tin Ka Ping P.A.T.H.S. Project” [26,27,28] to join the study. For Grade 7 students, 1362 and 1305 students joined in Wave 1 and Wave 2, respectively (attrition rate = 4.19%). For Grade 8 students, 1648 and 1343 students participated in the first year and second year, respectively (attrition rate = 18.51%). On both occasions, the students responded to the same questionnaire including measures of materialism, egocentrism and delinquency. Institutional ethics approval and consent from schools, parents and students were sought (HSEARS20161117001). When the data were collected, details of the study (including study purpose, data confidentiality and anonymity) were clearly explained to the students.

There were 2648 students in the final matched sample (1513 males and 1109 females, with no gender information for 26 students; mean age = 13.12 ± 0.81 years old). Parents of most respondents (N = 2225) were in their first marriage (i.e., intact families). In contrast, parents of 401 students were separated, divorced or re-married (i.e., non-intact families).

### 2.2. Measures

In both waves of data collection, a questionnaire assessing psychosocial development of adolescents in mainland China was used. In this study, we focused on the influence of materialism on adolescent delinquency via the mediating effect of egocentrism.

Assessment of materialism. Using the responses of 1658 adolescents, Shek, Ma and Lin [29] developed and validated the 21-item of the Chinese Adolescent Materialism Scale (CAMS), with reference conceptualizations and measures of materialism in Western cultures [2,3,4] as well as social beliefs of materialism in the local context. Although confirmatory factor analyses supported the original 4-factor model, there were only two items in the last factor. After deleting these two items, the 19-item measure showed very good psychometric properties. We used the 19-item measure in this paper. The first group of items surrounds the “centrality of acquiring material possession”. The items include: “I believe money is everything”; “my life goal is to earn as much money as possible”; “unless I can make a lot of money, I won’t respect myself”; “the most important objective in a woman’s life is to marry a rich man”; “the amount of money one makes is a fundamental indicator of one’s success”; “making money is more important than any other things”; “I believe the proverb that money makes the mare go”; and “people who own wealth own everything”. The second group of items is on the “value of material possession”: “possession of money can make people happy”; “people would love me if I am rich”; “the rich can get respect from others”; “possessing consumer goods brings joy to a person”; and “having the most updated smartphone makes one happy”. The final group of items is on hedonistic pursuits: “I believe that nothing goes well for a destitute couple”; “no money, no dignity”; “I would abandon some principles for the sake of money”; “I will not make friends with the poor”; “a life without money is meaningless to me”; and “one’s success rests with his/her wealth”. Reliability shows that the 19-item scale is internally consistent with Cronbach’s alpha as 0.94 and 0.95 at the two waves, respectively (see Table 1).

Assessment of egocentrism. Using 1658 adolescents as participants, Shek, Yu and Siu [30] developed the 19-item Chinese Adolescent Egocentrism Scale (CAES). As five items did not perform well, we eventually kept 14 items which showed that there are two dimensions. The finding supported the factor structure of the scale via confirmatory factor analysis (CFA), concurrent validity and reliability. The 14-item measure was used in this study. In the dimension on “self over others/disregard of others”, the items include: “I am loyal to my own feelings even if this may upset other people”; “it doesn’t matter how other people think”; “I agree that ‘every man for himself and the devil takes the hindmost’”; “my own benefits are more important than the benefits of other people”; “I often feel that I am more capable than people around me”; “my feelings are more important than the feelings of other people”; “no matter what happens, I can always justify my behaviors”; and “the criticisms on me are usually groundless”. For the dimension of “self-conceit”, the items are: “I am a unique person”; “I feel that fortune is always on my side”; “my views are often different from the views of other people”; “I believe that my views are superior to the views of other people”; “even though my ideas are different from others’, I would insist on mine”; and “my feelings are always different from the feelings of other people”. Reliability showed that the scale is internally consistent with Cronbach’s alpha as 0.85 and 0.86 at the two waves, respectively (see Table 1).

Assessment of Delinquency. We used 12 items to assess the occurrence of delinquent behaviors in the participants in the previous 12 months. These behaviors include stealing things, cheating at examinations, truancy, running away from home, damaging the properties of other people, attacking other people physically, having sex with others, fighting in gangs, speaking foul language, not returning home without parental permission, bullying or harassing other people and trespassing. For each item, there is a six-point scale (“0 = never, 1 = one to two times; 2 = three to four times; 3 = five to six times; 4 = seven to eight times; 5 = nine to ten times; 6 = more than ten times”). This scale has been validated and widely used among Chinese adolescents [31,32]. Reliability analyses showed that the scale was internally consistent in the present study (alpha as 0.81 and 0.86 at the two waves, respectively (see Table 1).

### 2.3. Analysis Plan

Consistent with the analytical approach adopted in previous studies [25,27,33,34], hierarchical multiple regression analyses were employed to examine the predictive effects of materialism and egocentrism on adolescent delinquency at each wave and over time. To examine the concurrent or longitudinal effects of materialism or egocentrism on adolescent delinquency, we entered the control variables (age, gender and family intactness) in the first block, followed by the relevant materialism or egocentrism variables. Furthermore, we examined whether Wave 1 materialism scores predicted change in Wave 2 delinquency scores, with control variables being entered in Step 1, Wave 1 delinquency scores entered in Step 2 and Wave 1 materialism scores entered in Step 3. As Steinberg, Elmen and Mounts [35] pointed out, “despite recent advances in structural equation modeling, it is still generally agreed that the use of multiple regression techniques in which one predicts scores on a dependent variable at time 2 while controlling for scores on that same variable at time 1 is an appropriately a conservative strategy” (p. 1428). To examine the mediating effect of egocentrism on the influence of materialism on adolescent delinquency, we conducted PROCESS analyses to examine the indirect effect of Wave 1 materialism on Wave 2 delinquency via Wave 2 egocentrism as a mediator [36].

## 3. Results

### 3.1. Attrition Analyses

As far as sample attrition over time is concerned, results showed that student dropout at Wave 2 was not high. Besides, the background socio-demographic characteristics of those who dropped out of the study and those who did not drop out were basically similar. Furthermore, the matched sample and the dropouts did not have a significant difference regarding their delinquency and materialism levels at Wave 1. The only significant difference was that egocentrism was lower in the study sample than the dropouts at Wave 1 (*t* = 4.40, *p* < 0.001, Cohen’s *d* = 0.27).

### 3.2. Pearson Correlation Analyses

Concurrent and longitudinal correlation coefficients on the inter-relationships amongst materialism, egocentrism and adolescent delinquency are shown in Table 2. While materialism and egocentrism were positively correlated with adolescent delinquency (*r* = 0.26 and 0.14, *p* < 0.001, respectively), materialism had a positive relationship with egocentrism at Wave 1 (*r* = 0.45, *p* < 0.001). Similar observations were found for Wave 2 (materialism and delinquency: *r* = 0.27, *p* < 0.001; egocentrism and delinquency: *r* = 0.17, *p* < 0.001; materialism and egocentrism: *r* = 0.48, *p* < 0.001). Results also showed that Wave 1 materialism and egocentrism were positively associated with Wave 2 adolescent delinquency scores (*r* = 0.17 and 0.08, *p* < 0.001, respectively).

### 3.3. Concurrent and Longitudinal Multiple Regression Analyses

For all multiple regression analyses, the control variables (age, gender and family intactness) were entered at Step 1, followed by the relevant predictor variable(s). For Wave 1 analyses, Table 3 shows that materialism and egocentrism separately predicted delinquency (*β* = 0.25, *p* < 0.001, Cohen’s *f*^2^ = 0.06 and *β* = 0.11, *p* < 0.001, Cohen’s *f*^2^ = 0.02, respectively). Similarly, Wave 2 materialism and egocentrism predicted Wave 2 delinquency (*β* = 0.25, *p* < 0.001, Cohen’s *f*^2^ = 0.07 and *β* = 0.16, *p* < 0.001, Cohen’s *f*^2^ = 0.03, respectively). For longitudinal prediction analyses shown in Table 4, Wave 1 materialism and Wave 1 egocentrism separately showed significant prediction of Wave 2 delinquency (*β* = 0.15, *p* < 0.001, Cohen’s f2 = 0.02 and β = 0.07, *p* < 0.01, Cohen’s f2 = 0.01, respectively). These findings provided support for Hypotheses 1a, 1b, 2a, 2b, 3a and 3b.

To examine the influence of materialism or egocentrism on the change in delinquency over time, after entering the control variables at Step 1 and Wave 1 delinquency scores at Step 2, Wave 1 materialism or egocentrism scores were included in Step 3. Results showed that while Wave 1 materialism positively predicted increase in Wave 2 delinquency (*β* = 0.05, *p* < 0.01, Cohen’s *f*^2^ = 0.003), egocentrism at Wave 1 did not predict change in delinquency at Wave 2 (*β* = 0.02, *p* > 0.05).

### 3.4. Mediating Effect

Results on the mediating analyses can be seen in Table 5. PROCESS analyses with 5000 bootstrapping showed that egocentrism at Wave 2 mediated the influence of Wave 1 materialism on Wave 2 delinquency with a small effect size (mediating effect = 0.02, *p* < 0.001, bias-corrected 95% confidence interval = [0.01, 0.03]). The results provided support for Hypothesis 4.

## 4. Discussion

There are several constructive responses to the methodological problems outlined in the Introduction. First, in view of the paucity of related research findings based on adolescents, we employed high school students as participants in this study. Second, as there are very few Chinese studies in this field, we collected data from mainland Chinese high schools. Third, we used validated Chinese measures in this study [29,30,32]. Finally, we conducted a short-term longitudinal study to examine the relationships amongst materialism, egocentrism and adolescent delinquency. With reference to the methodological weaknesses of the studies in the literature [5], these features are methodological advances in the research design.

Besides, we also addressed two conceptual issues discussed above in this paper. First, as there are few research studies examining materialism and antisocial behavior, the present study examined this relationship. Computer search based on PsycINFO on 6 September 2020 showed that there were 20 citations when we used “materialism” and “antisocial behavior” (or delinquency) as the search terms. Second, although Donnelly et al. [6] proposed that materialism may adjust self-construction, which may in turn serve as a mediator linking materialism and adolescent antisocial behaviors, there is no coverage on the linkages among materialism, egocentrism, self-centeredness and/or narcissism and adolescent delinquency. Thus, we examined the influence of materialism on adolescent antisocial behaviors via egocentrism. This is important because the related findings can enrich our understanding of the dynamic relationships amongst materialism, egocentrism and antisocial behaviors. Actually, when we used the search terms of “materialism”, “egocentrism” and “antisocial behavior” on 6 September 2020, PsycINFO revealed only one citation.

Consistent with our expectation, materialism significantly predicted adolescent delinquency at each wave. Materialism at Wave 1 also predicted delinquency and its change at Wave 2. Although the effect size of the findings is small, these novel findings suggest that apart from those frequently used indicators of well-being (e.g., life satisfaction) [5], materialism also predicts adolescent delinquency. Conceptually speaking, it expands the conceptual model of the influence of materialism on adolescent developmental outcomes including antisocial behavior. The present findings also enrich the scientific literature that materialism is a risk factor for adolescent antisocial behaviors. Moreover, the pioneering and significant findings will provide insights and empirical support to the prevention and intervention programs coping with delinquent behaviors of adolescents with materialistic values.

Regarding the influence of egocentrism on delinquency, the findings are also consistent with our expectation and the findings that egocentrism [24] and lack of empathy [23] are positively related to adolescent delinquency. However, although egocentrism predicted delinquency at each wave and over time, Wave 1 egocentrism did not predict change in delinquency at Wave 2. This observation suggests that there is a need to replicate the present findings. As egocentrism represents a self-focused tendency [6] where one regards his/her interest to be more important than the interest of other people, the present findings expand our understanding of the role of self-representations (i.e., egocentrism) on antisocial behaviors. The findings also suggest that egocentrism is a risk factor of adolescent delinquency.

Besides, we also found that materialism predicted egocentrism at Wave 1 and Wave 2. Over time, Wave 1 materialism also predicted Wave 2 egocentrism and its change, although the effect size was small. The findings expand the theoretical formulation of Donnelly et al. [6] that materialism leads to self-deconstruction that includes egocentrism. Theoretically, self-deconstruction has narrow attention which focuses on the present. This feature is consistent with egocentrism where a person focuses on himself/herself and exaggerates the importance of oneself. As few studies have examined the influence of materialism on egocentrism, this is a pioneer study in the literature. In the future, studies should explore whether materialism would have a positive correlation with egocentrism over time.

Finally, the findings provide support for the hypothesis that egocentrism is a mediator of the influence of materialism on delinquency. There are two contributions of the present findings. First, they underscore the dynamic relationships amongst materialism, egocentrism and adolescent antisocial behaviors. Second, they demonstrate the important role of self-representation (as indexed by egocentrism in this study) regarding the impact of materialism on delinquency. In the future, as there are several “steps” in the model of Donnelly et al. [6], longitudinal studies should be conducted to examine how self-representation as a result of materialism would lead to adolescent antisocial behavior over time.

In addition to the theoretical implications mentioned above, the present findings have two practical implications. First, changing materialistic beliefs of adolescents would help to reduce adolescent antisocial behavior. With reference to the major theories in the positive youth development (PYD) literature, helping adolescents to develop positive values is an important adolescent developmental task. Positive values are an important internal asset in the model of developmental assets [37]. Character is emphasized in the “5Cs model” (connection, competence, confidence, character and caring) proposed by Lerner et al. [38]. In the model of Peterson and Seligman [39], it is proposed that character strength is important. Second, the present findings suggest that the reduction of egocentrism would help to reduce adolescent delinquency. In the PYD literature, cognitive competence is important. In the “5Cs model”, competence is one of the factors leading to positive adolescent development. In the social-emotional learning literature, there is an emphasis on empathy and social awareness in promoting adolescent thriving [40]. There are PYD programs worldwide showing that the promotion of PYD attributes can help to reduce adolescent delinquency [28,41,42,43]. There is also evidence showing that different stakeholders regard cultivation of PYD attributes in adolescents as important [44].

Although this study is pioneering in this field, it has several limitations. First, we should realize the limitation of a short-term longitudinal study which has only two time points. Ideally speaking, we should design a longitudinal study with more time points which can help us further understand the long-term inter-relationships amongst materialism, egocentrism and adolescent delinquency. This is particularly important when we test the framework proposed by Donnelly et al. [6] that there are several stages of self-appraisal triggered by materialism. Second, as we only collected data from four schools, generalizability of the present findings should be made with caution. Third, besides self-report data from adolescents, data collected from significant others, such as parents, would help to triangulate the findings. Besides, in addition to quantitative studies, qualitative studies can help to enrich our understanding of the dynamic relationships amongst these three constructs. Case studies of the related behavior in delinquents would also be illuminating. Finally, as the effect sizes of the findings surrounding the multiple regression analyses are not high, there is a need to realize the distinction between statistical significance and practical significance. Obviously, efforts to replicate the present findings to consolidate the related observations are important. Nevertheless, in view of the few longitudinal Chinese studies examining egocentrism as a mediator in the relationship between materialism and delinquency, the present study offers theoretical insights into the role of materialism and egocentrism in adolescent delinquency based on Chinese adolescents in mainland China.

## 5. Conclusions

The present study addressed several conceptual and methodological limitations in the scientific literature on the association between materialism and well-being. Conceptually, we examined the influence of materialism on well-being indexed by adolescent delinquency, investigated the influence of materialism on egocentrism and tested egocentrism as a mediator of the linkage between materialism and adolescent delinquency. Methodologically, we recruited mainland Chinese high school students and used validated measures in a short-term longitudinal study with a large sample. Consistent with our expectations, the present study showed that materialism positively predicted egocentrism and adolescent delinquency over time, with egocentrism serving as a mediator. In the future, we will further examine the findings of this study in other cultures, in particular, we would like to explore whether the cultural differences between collective and individualistic societies would influence the impacts of materialism and egocentrism on adolescent delinquency.

## Figures and Tables

**Table 1 ijerph-17-07662-t001:** Reliability of measures.

Measures	Wave 1		Wave 2	
	α	Mean Inter-Item Correlation	α	Mean Inter-Item Correlation
Materialism	0.94	0.47	0.95	0.50
Egocentrism	0.85	0.31	0.86	0.32
Delinquency	0.81	0.26	0.86	0.34

**Table 2 ijerph-17-07662-t002:** Descriptive and correlational analyses.

Measures	Mean	SD	Correlations							
			1	2	3	4	5	6	7	8
1. Age	13.12	0.81	--							
2. Gender ^a^			−0.08 ***	--						
3. Family intactness ^b^			0.03	0.02	--					
4. W1 MT	2.22	0.99	0.14 ***	−0.14 ***	0.06 **	--				
5. W1 EG	3.13	0.90	0.05**	−0.13 ***	0.02	0.45 ***	--			
6. W1 DE	0.45	0.52	0.11***	−0.14 ***	0.08 ***	0.26 ***	0.14 ***	--		
7. W2 MT	2.44	1.08	0.16 ***	−0.12 ***	0.05 **	0.48 ***	0.23 ***	0.22 ***	--	
8. W2 EG	3.20	0.89	0.09 ***	−0.15 ***	0.02	0.23 ***	0.30 ***	0.12 ***	0.48 ***	--
9. W2 DE	0.42	0.57	0.07 ***	−0.12 ***	0.07 ***	0.17 ***	0.08 ***	0.43 ***	0.27 ***	0.17 ***

*Note.*^a^ 1 = male, 2 = female; ^b^ 1 = intact, 2 = non-intact. W1 = Wave 1; W2 = Wave 2; MT = materialism; EG = egocentrism; DE = delinquency. ** *p* < 0.01; *** *p* < 0.001.

**Table 3 ijerph-17-07662-t003:** Cross-sectional regression analyses for delinquency.

Model	Predictors	Delinquency (Wave 1)					Delinquency (Wave 2)				
		*β*	*t*	Cohen’s *f*^2^	R^2^ Change	F Change	*β*	*t*	Cohen’s *f*^2^	R^2^ Change	F Change
1	Age	0.04	2.52 *	0.001	0.027	35.84 ***	0.06	2.95 **	0.003	0.022	19.10 ***
	Gender ^a^	−0.12	−7.52 ***	0.019			−0.11	−5.65 ***	0.012		
	Family intactness ^b^	0.10	6.27 ***	0.007			0.07	3.58 ***	0.005		
2	Age	0.01	0.85	0.005	0.061	256.13 ***	0.02	1.07	0.000	0.061	171.15 ***
	Gender ^a^	−0.09	−5.85 ***	0.011			−0.08	−4.36 ***	0.007		
	Family intactness ^b^	0.09	5.50	0.005			0.06	3.12 **	0.004		
	Materialism	0.25	16.00 ***	0.060			0.25	13.08 ***	0.067		
2	Age	0.04	2.26 *	0.009	0.013	50.61 ***	0.05	2.37 *	0.002	0.024	64.17 ***
	Gender ^a^	−0.11	−6.60 ***	0.014			−0.09	−4.45 ***	0.008		
	Family intactness ^b^	0.10	6.11 ***	0.006			0.06	3.33 **	0.004		
	Egocentrism	0.11	7.11 ***	0.015			0.16	8.01 ***	0.025		

*Note.* Measures of materialism and egocentrism at Wave 1 and Wave 2 were included as predictors to predict delinquency at Wave 1 and Wave 2, respectively. ^a^ 1 = male, 2 = female; ^b^ 1 = intact, 2 = non-intact. * *p* < 0.05; ** *p* < 0.01; *** *p* < 0.001.

**Table 4 ijerph-17-07662-t004:** Longitudinal regression analyses for delinquency.

Model	Predictors	Delinquency (Wave 2)					Delinquency (Wave 2)				
		*β*	*t*	Cohen’s *f*^2^	R^2^ Change	F Change	*β*	*t*	Cohen’s *f*^2^	R^2^ Change	F Change
1	Age	0.06	3.20 **	0.004	0.021	18.58 ***	0.02	1.32	0.001	0.167	527.91 ***
	Gender ^a^	−0.11	−5.58 ***	0.012			−0.05	−2.96 **	0.003		
	Family intactness ^b^	0.06	3.23 **	0.004			0.03	1.66	0.001		
	Wave 1 Delinquency						0.42	22.98 ***	0.206		
	Age	0.04	2.24 *	0.002	0.022	58.49 ***	0.02	1.00	0.000	0.003	8.24 ***
	Gender ^a^	−0.09	−4.59 ***	0.008			−0.05	−2.66 **	0.003		
2	Family intactness ^b^	0.05	2.79 **	0.003			0.03	1.55	0.001		
	Wave1 Delinquency						0.40	21.65 ***	0.183		
	Wave1 Materialism	0.15	7.65 ***	0.023			0.05	2.87 **	0.003		
	Age	0.06	3.01 **	0.003	0.004	11.46 ***	0.02	1.26	0.001	0.000	0.67
	Gender ^a^	−0.10	−5.12 ***	0.010			−0.05	−2.88 **	0.003		
2	Family intactness ^b^	0.06	3.11 **	0.004			0.03	1.63	0.001		
	Wave1 Delinquency						0.41	22.69 ***	0.201		
	Wave1 Egocentrism	0.07	3.39 **	0.005			0.02	0.82	0.000		

*Note.*^a^ 1 = male, 2 = female; ^b^ 1 = intact, 2 = non-intact. * *p* < 0.05; ** *p* < 0.01; *** *p* < 0.001.

**Table 5 ijerph-17-07662-t005:** Mediating effect of Wave 2 egocentrism (the mediator) for the effect of Wave 1 materialism on Wave 2 delinquency.

Regression Models Summary	*B*	*SE*	*t*
Total effect of Wave 1 materialism (IV) on Wave 2 delinquency (DV)	0.08	0.01	7.54 ***
Wave 1 materialism (IV) to Wave 2 egocentrism (Mediator)	0.19	0.02	10.87 ***
Wave 2 egocentrism (Mediator) to Wave 2 delinquency (DV)	0.08	0.01	6.66 ***
Direct effect of Wave 1 materialism (IV) on Wave 2 delinquency (DV)	0.07	0.01	6.03 ***
Mediating effect of Wave 2 egocentrism (Mediator)	**Point estimate**	**Bootstrapping** **(BC 95% CI)**	
		**Lower**	**Upper**
	0.02 ***	0.01	0.03

*Note*. In all analyses, control variables were statistically controlled. IV = independent variable; DV = dependent variable; BC = bias-corrected; CI = confidence interval. *** *p* < 0.001.

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
