# Peer review of "Concurrent and Longitudinal Predictors of Adolescent Delinquency in Mainland Chinese Adolescents: The Role of Materialism and Egocentrism"

_ijerph, 2020, doi:10.3390/ijerph17207662_

Round 1

Reviewer 1 Report

This study wished to test hypotheses relating to the influence of materialism and egocentrism on adolescent delinquency.  It improves upon earlier work by using a much larger sample, youth from mainland China (which has about 19% of the world's population), a two point in time longitudinal design. The data were obtained from the Project PATHs project in mainland China.  With this design it was possible to look for correlations among the variables at time 1, and again at time 2.  And also to use the time one data to predict delinquency at time 2. Attrition at Time 2 was remarkably low, and overall this study was a solid advance in research methodology. Psychometrically valid measures were used to assess materialism and ego centrism, measures developed in a Chinese context.

Table 1 has the top line of numbers misaligned to the right.  They should be properly adjusted to the left.

While the hypotheses were 'supported' in the sense that significant correlations were observed, the correlations were often rather small, and the associated r-squared (the coefficient of determination) are even smaller.  So while the H were supported, the practical meaningfulness of the correlations is very small. This think that this should be mentioned more in the discussion. The presentation of the limitations is well done.

In summary this is well designed and well written.  I hope it is accepted by the Editor.

Author Response

Please refer to the attached response letter. Many thanks. 

Reviewer 2 Report

The article is interesting and well-written. I am impressed by the size of the sample and rigorously followed methodological standards. However, despite a generally positive evaluation of the article, I would like to point out some weak points.

First, the introductory part of the article is too long and contains fragments that seem to be not relevant to the main topic. The authors pay a lot of attention to the relationship between materialism and well-being and seem to treat juvenile delinquency as an aspect of well-being. Such an approach needs to be better explained.

Second, the authors do not mention at all probably the most influential in social sciences   – alongside the concept of Kasser and Ryan – conceptualisation of materialism proposed by Richins and Dawson (1992; see also Richins, 2004) and their Material Values Scale and define materialism following the marginal concept by Russell W. Belk. At the same time they develop their own scale with factors corresponding with aspects of materialism listed by Richins and Dawson (at least two of them – centrality and possession-related happiness).

Third, I am not sure if the authors’ reading of the article by Donnelly et al. (2016) is accurate. My reading of the “Escaping from the self theory of materialism”, presented by Donnelly et al.,  is different. In my opinion Donnelly et al. present a process which results in materialistic behaviour and attitudes rather than a process which arises from materialism. This way of thinking follows Roy Baumeister’s idea that the process of escaping from self-awareness may explain various dysfunctional behaviors in people.  However, Donnelly and his co-workers in their article tend to avoid specifying the direction of the relationships between phenomena appearing in the six steps of the self-escape process. Donnelly et al. (2016) write:  “The six steps of escape theory constituted the current framework, which furnished the basis for hypothesizing that materialists would exhibit the following patterns more than other people (i.e., more than nonmaterialists). First, materialists should tend to be characterized by falling short of standards and feeling disappointed with their life. Second, they should be disposed to blame themselves for their setbacks, stresses, disappointments, perceived inadequate social standing, and other undesirable outcomes. Third, materialists should be prone to experience high self-awareness. Fourth, dysphoric moods should be common. Fifth, materialists should show signs of cognitive deconstruction, such as narrow, rigid, concrete, and present-focused thinking, which would aid escape from aversive moods and emotions. Last, materialism should be associated with impulsive, short-sighted, and irrational behavior patterns. There may also be attempts to embrace a new, different version of the self, which would complete the escape from the former self” (p.299; underlining by the reviewer). However, the authors of the reviewed manuscript claim that materialism influences (“leads to”) these phenomena (e.g. line 168-169, 208-209). In my opinion, it is theoretically incorrect.

I am also not convinced that introducing egocentrism to the research model might be justified by the self-escape theory of materialism. Where is a place for egocentrism in this model?

There are also some methodological questions I would like to ask. First, why the authors constructed their own tools for measuring materialism and egocentrism? The issue needs further explanation. Second, why materialism and egocentrism were entered separately to different regression models? It seems obvious that they should be examined as predictors of delinquency together.

And the last remark - the study was conducted on the Chinese sample. Do authors think that the results would be different while obtained from youngsters from different cultures? Does the collectivism/individualism cultural dimension play any role in the relationship between materialism, egocentrism and juvenile delinquency?

Author Response

(The authors gave the same response as above.)
